# Corrosion Behavior of the 2024 Aluminum Alloy in the Atmospheric Environment of the South China Sea Islands

Jing Zhao [1], Tongjun Zhao [2], Yazhou Zhang [2], Zhongtian Zhang [2], Zehao Chen [2,*], Jinlong Wang [2] and Minghui Chen [2]

[1] Nuclear Power Institute of China, Chengdu 528208, China
[2] Shenyang National Laboratory for Materials Science, Northeastern University, Shenyang 110819, China; 15040646273@163.com (T.Z.); 2210262@stu.neu.edu.cn (Y.Z.); 15633570693@163.com (Z.Z.); wangjinlong@mail.neu.edu.cn (J.W.); mhchen@mail.neu.edu.cn (M.C.)
* Correspondence: chenzehao@mail.neu.edu.cn; Tel.: +86-24-23904856; Fax: +86-24-23893624

**Abstract:** The 2024 aluminum alloy, a structural material commonly used in aviation aircraft bodies, is susceptible to serious corrosion in marine atmospheric environments. This paper comprehensively studies the corrosion behavior of the 2024 aluminum alloy in the South China Sea atmosphere. Weighing, morphology observation, phase analysis, electrochemical testing, and other methods were used to study the corrosion law and corrosion mechanism of the 2024 aluminum alloy. The main conclusions are as follows: At the initial stage of exposure, pitting corrosion occurred on the surface of the 2024 aluminum alloy. After 3 months of exposure, the self-corrosion current density increased from 0.456 $\mu A \cdot cm^{-2}$ to 8.338 $\mu A \cdot cm^{-2}$. After 6 months of exposure, the corrosion developed into general corrosion. The main component of the corrosion product was $Al_2O_3 \cdot 3H_2O$. The product covered the surface to form a loose corrosion product layer, which had an inhibitory effect on corrosion. The self-corrosion current density was reduced to 2.359 $\mu A \cdot cm^{-2}$. After 12 months of exposure, the corrosion product layer fell off and became thinner, and the self-corrosion current density increased to 2.849 $\mu A \cdot cm^{-2}$. The corrosion kinetics conformed to the functional equation $W = 0.00346t^{0.73891}$, indicating that the corrosion products have a certain protective effect on the matrix.

**Keywords:** 2024 aluminum alloy; corrosion product; pitting corrosion; marine atmosphere

## 1. Introduction

Aluminum alloy is widely used as a structural material in the transportation field due to its low density, favorable plasticity, and good corrosion resistance, especially high-strength aluminum alloy, which is the main profile for manufacturing high-speed rail car bodies and aircraft fuselages [1–4]. In recent years, the atmospheric environment, faced by high-speed railways, airplanes, and other vehicles, has become increasingly complex as the scope of construction of transportation facilities has become wider. Because aluminum atoms have a relatively strong affinity for oxygen, self-passivation of aluminum alloy can occur at room temperature, forming an oxide film with a thickness of several nanometers on the surface [5–7]. Thus, in a general atmosphere, aluminum alloys are well resistant to corrosion. However, in the marine atmosphere with high chloride ion content, aluminum alloys are susceptible to corrosion, such as pitting corrosion, intergranular corrosion, exfoliation, etc. [8], resulting in a serious decline in the mechanical properties of aluminum alloys [9]. The service life of aluminum alloys is significantly shortened or even fails prematurely. Therefore, the corrosion behavior of aluminum alloys in the marine atmosphere has always been an important research direction [10–12].

Commonly, there are two types of atmospheric corrosion research methods: field exposure and indoor accelerated testing [10,13–15]. The field exposure method involves placing the samples in a realistic environment for experimentation; so, the data obtained is very reliable. It has been an influential method for studying atmospheric corrosion.

Zhao Qiyue et al. [15] studied the corrosion behavior of the 7A85 aluminum alloy exposed to the industrial-marine atmosphere for 5 years. They found that intergranular corrosion occurred along the second phase of the alloy, and the mechanical properties of the aluminum alloy decreased significantly.

The 2024 aluminum alloy belongs to the Al-Cu-Mg series of alloys, has a fine second phase distributed internally, and is high-strength duralumin. Due to its good plasticity, it is often used to make various high-load parts and components, such as aircraft skins, spars, ribs, etc. It has been widely used as a structural material in civil and military aircraft and is an indispensable and essential structural material in the aviation industry, but its corrosion resistance is not universal. Marie-Laetitiade Bon-fils-Lahovary et al. [16] found that aluminum in the 2024 aluminum alloy and the second phase forms a galvanic cell, and an electrochemical reaction occurs in the corrosive medium to cause intergranular corrosion. Zhang Sheng et al. [10] found that after exposure to the marine atmosphere of the 2024-T4 aluminum alloy for 7 years, severe pitting and intergranular corrosion occurred in the aluminum alloy. After 20 years of exposure, the aluminum alloy was exfoliated and the main corrosion product was $Al(OH)_3$.

The South China Sea has extremely harsh corrosive environments because the average temperature is 27 °C and the highest temperatures exceed 35 °C. The average value of relative humidity (RH) is 77%, and the highest humidity is 85% [17–19]. The corrosivity of the environment is CX, according to ISO 9223-2012 standard [20]. There are few reports on the corrosion behavior of the 2024 aluminum alloy in the atmosphere of the South China Sea using on-site exposure methods. Therefore, this paper presents a 12-month field exposure experiment on the 2024 aluminum alloy in the atmospheric environment of the South China Sea. The corrosion behavior of the 2024 aluminum alloy was characterized by weighing, morphological observations, phase identification, electrochemical testing, and other analytical methods, which provided data to support the corrosion protection design of the aircraft fuselage and maintenance methods for the aircraft when parked. This study offers new ideas and is therefore of crucial practical importance.

## 2. Materials and Methods

### 2.1. Materials

The samples used in the atmospheric corrosion test were processed by the Commercial Aircraft Corporation of China. The dimensions of the samples were 150 mm × 75 mm × 5 mm. The surface of the samples was refined with a roughness of Ra6.0. Then, samples were placed into a mixture of ethanol and acetone (volume ratio 3:1), cleaned with an ultrasonic cleaner to remove oil stains on the surface, and then the time was set to 30 min. The chemical composition of the 2024 aluminum alloy is listed in Table 1. Environmental characteristic parameters of the Xisha Islands are shown in Table 2.

**Table 1.** The chemical composition of the 2024 aluminum alloy (wt.%) [12,21,22].

| Element | Cu | Mg | Mn | Cr | Zn | Al |
|---------|------|------|-------|-----|------|------|
| Content | 3.8–4.9 | 1.2–1.8 | 0.3–1.0 | 0.1 | 0.25 | Bal. |

**Table 2.** Environmental characteristic parameters of the Xisha Islands.

| Average Temperature (°C) | Average Humidity (%) | Sunshine Duration (h/a) | Total Rainfall (mm/a) | Surface Wetting Time (h/a) | $Cl^-$ Deposition Rate (mg/100 cm²·d) |
|---|---|---|---|---|---|
| 26–27 | 80 | 2800 | 1500 | 2628 | 1.0 |

### 2.2. Atmospheric Corrosion Test

First, the samples were numbered (01–36) by using an ink signature pen, and the numbered side was defined as the reverse side. Then, the samples were weighed by the electronic balance (SECURA225D-1CN, Sartorius, German with the precision of 0.1 mg)

and measured by the vernier caliper (precision 0.01 mm) to obtain the length, width, height, and weight. Then, a photo of the front side of the samples was taken. The samples were packaged and transported to the Paracel Islands for the exposure test. Finally, the samples were firmly installed on the exposure test frame and then inclined at 45° to the horizontal surface. The lengths of exposure time were 1, 2, and 3 weeks and 1, 3, 6, and 12 months, and 3 parallel samples were retrieved each time. After retrieving the samples, the surface of the samples was washed with distilled water and dried in an oven. After being dried, the samples were weighed again. The added weight of the samples was calculated according to the following formula:

$$W = \frac{m_1 - m_0}{2(h \times k + h \times l + k \times l)} \tag{1}$$

In the formula, W is the weight gain per unit area (mg·cm$^{-2}$), $m_1$ is the weight when the samples were retrieved (mg), $m_0$ is the original weight of the samples (mg), and $h, k, l$ are the height, width, and length (cm) of the samples, respectively.

### 2.3. Morphology Observation

In this paper, the backscattered electron mode was used to analyze the cross-sectional micromorphology. A scanning electron microscope (SEM, Inspect F50, FEI Co., Ltd., Hillsboro, OR, USA) with an accelerating voltage of 20 kV was used.

### 2.4. Phase Analysis

The X-ray diffractometer (XRD, CuKα radiation at 40 kV, 30 mA, SmartLab, Rigaku, Japan) was used in this study. Cu was used as the target, the acceleration voltage was 40 kV, and the 2θ range was 10° to 90°, with a step of 0.02°/point.

### 2.5. Electrochemical Test

2.5.1. AC Impedance Test (EIS)

The test adopted a three-electrode system (CHI660E, ChenHua, Shanghai, China), using a saturated calomel electrode as a reference electrode and a Pt electrode as an auxiliary electrode. The test solution was 3.5 wt.% NaCl solution. The test solution was not deaerated. Before the AC impedance test, we performed an open circuit potential test first and then performed AC after the open circuit potential was stable. For the impedance test, the test frequency range was 0.01 Hz~100 kHz and the disturbance voltage amplitude was 5 mV.

2.5.2. Polarization Test

After the AC impedance test, the samples were tested for the potential polarization curve (CHI660E, ChenHua, China), and the test solution was 3.5 wt.% NaCl solution. The test solution was also not deaerated, which adopted a three-electrode system, with a scanning range of −0.3 V (vs. E$_{ocp}$) to +0.6 V (vs. E$_{ocp}$) and a scanning speed of 1 mV/s.

## 3. Results and Discussion

### 3.1. Corrosion Kinetics

The relationship between the weight gain per unit area of the samples in the outdoor environment and the exposure time is shown in Figure 1. The figure shows that the weight gain per unit area of the samples gradually increases with the prolongation of the exposure time. This is due to the accumulation of corrosion products on the surface of the samples. In the early exposure stage, the corrosion weight gain rate is relatively fast with the extension of the exposure time. In the later stage, the corrosion weight gain rate begins to slow down. When the exposure time reaches 12 months, the corrosion weight gain per unit area is 1.656 mg·cm$^{-2}$. Formula (2) was used to fit the weight gain scatter points, where A and n are constants, and the fitting curve is shown in Figure 1, where the value of A is 0.00202. The value of n is 0.7389, and the fitted correlation coefficient is 0.98995, indicating that the fitting is reasonable. According to related literature, the power exponent n value in the power function of corrosion kinetics is affected by material characteristics and environmental

factors. When the value of n is less than 1, the corrosion product has a protective effect on the matrix; when the value of n is greater than 1, the corrosion product does not have protection. The corrosion kinetic power exponent value of the samples in the outdoor environment is 0.73891 because the value of n is less than 1; therefore, from the perspective of corrosion kinetics, in the outdoor environment, the corrosion products have a certain protective effect on the matrix.

$$W = At^n \tag{2}$$

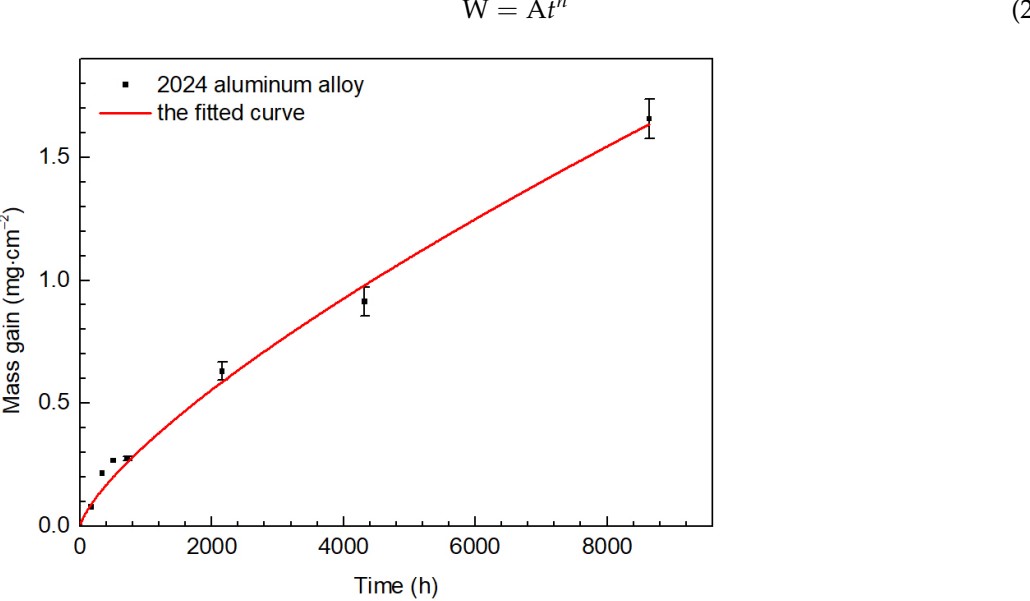

**Figure 1.** The relationship between the weight gain per unit area of the samples in the outdoor environment and the exposure time.

### 3.2. Surface Morphologies

The front macroscopic appearances of the samples with different exposure time lengths are shown in Figure 2. The figure shows that the surface of the samples is very bright and has a metallic luster because of the fine lathe processing. After outdoor exposure for 1 week, the surface of the samples had scattered white pitting corrosion products attached, but the area was relatively small. Hence, the surface of the samples was still bright and still had a metallic luster. After 2 weeks of outdoor exposure, the number of white spots and the coverage area increased. Over time, more white corrosion products occurred on the surface of the samples, and the coverage area also began to increase. After 3 months of exposure, the surface of the samples was no longer bright and completely lost its metallic luster. The initial scattered dots on the surface of the samples also turned into a thin layer of flakes. When the exposure time reached 6 months, the color of the samples' surface deepened, indicating that the corrosion products further accumulated on the surface of the samples and completely covered the test surface. When the exposure time reached 12 months, the surface of the samples was entirely occupied by corrosion products, and the corrosion product layer cracked, resulting in an uneven surface of the samples. Therefore, the samples were darker in color.

The surface microstructure morphologies of the samples are shown in Figure 3, where the red letters (a, b, c, d and e) in the figure are the scanning area of the energy spectrum. After 1 week of exposure, scattered pitting corrosion appeared on the surface of the samples, but the diameter of the pit was very small, about 14 μm, and there were corrosion products in the pits. After 2 weeks of exposure, the diameter of the pits expanded to 20.5 μm. After 3 weeks of exposure, the diameter of the pits expanded further, and some of the pits were connected to form a relatively large pitting area. When the exposure time reached 1 month, the area where the pitting corrosion occurred was further enlarged, and at the same time, the corrosion products adhered to the pitting area.

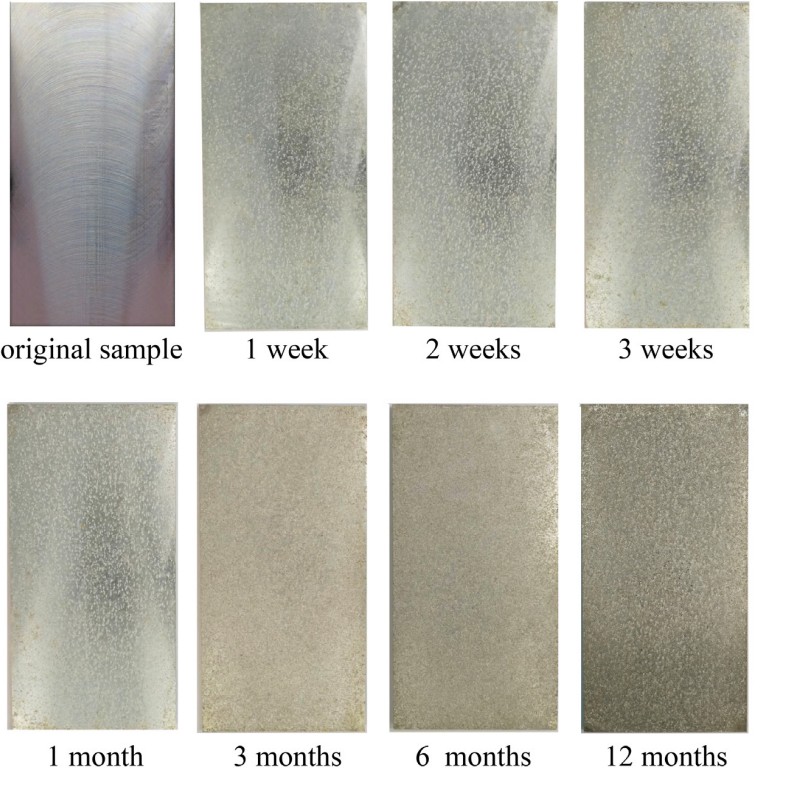

original sample     1 week     2 weeks     3 weeks

1 month     3 months     6 months     12 months     50mm

**Figure 2.** The front macroscopic appearance of the samples with different exposure time lengths.

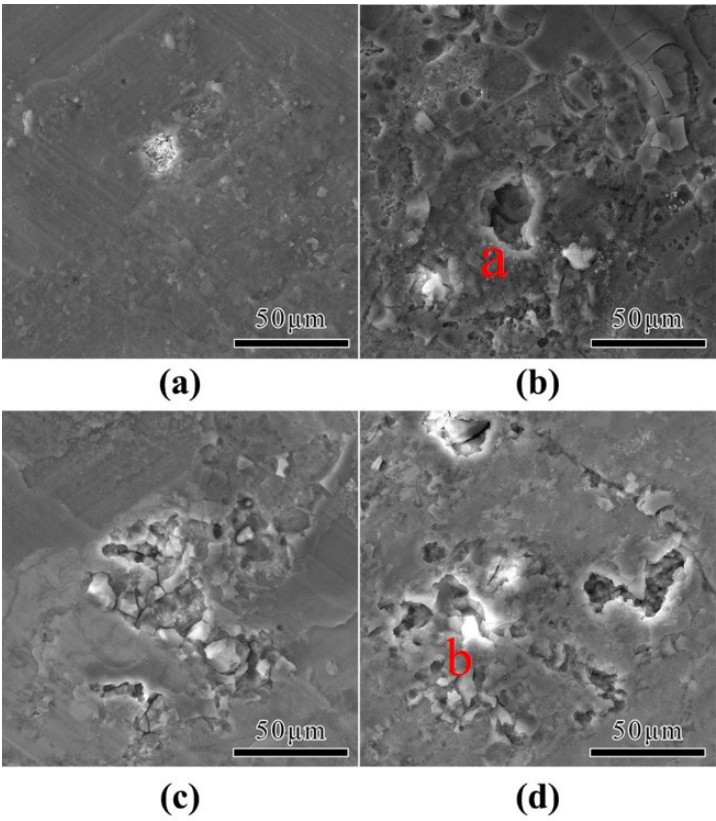

**Figure 3.** The surface microstructure morphologies of the samples with different exposure time lengths: (**a**) 1 week, (**b**) 2 weeks, (**c**) 3 weeks, and (**d**) 1 month.

The cross-sectional morphologies of the samples with different exposure time lengths are shown in Figure 4. The white particles in the figure are the second phase in the aluminum alloy. The figure shows that pitting corrosion appeared on the surface of the samples after 1 week of exposure. Then, after 2 weeks of exposure, the distribution density of pits increased, and the distance between pits was relatively short. After reaching 3 weeks, the diameter of the pits became larger and some pits tended to be connected to each other. When the exposure time reached one month, due to the increase in the number of pits and the enlargement of their diameters, some pits were connected to form a larger area of pitting corrosion, and the gray areas in the figure are corrosion products. It can be seen that the corrosion products are in the spots. Accumulation appeared in the pits and located near the holes.

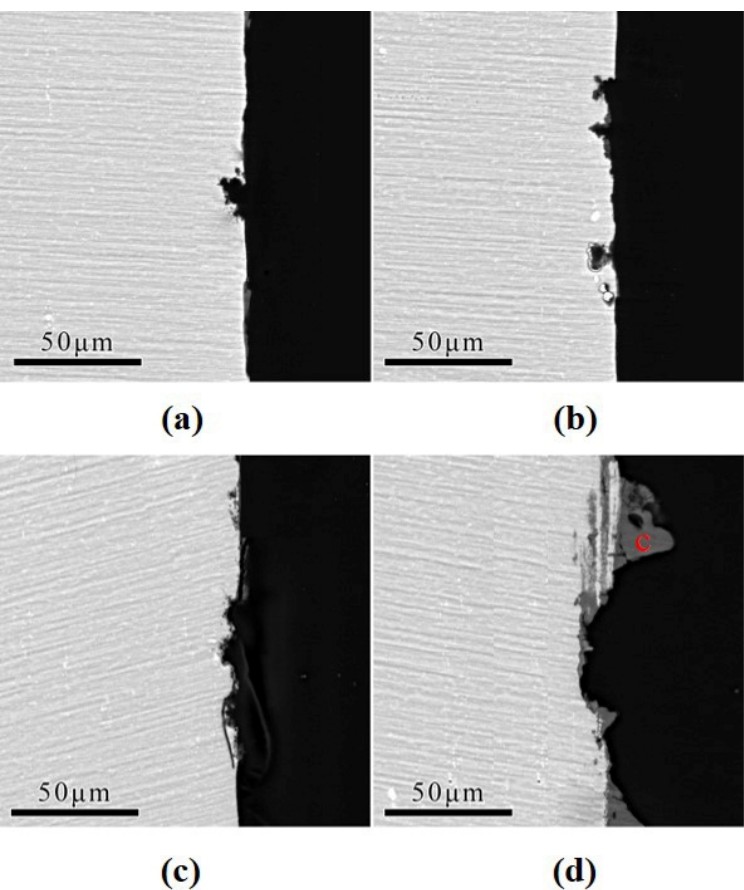

**Figure 4.** The cross-section microstructure morphologies of the samples with different exposure time lengths: (**a**) 1 week, (**b**) 2 weeks, (**c**) 3 weeks, and (**d**) 1 month.

More corrosion products accumulated, and the convex area thickened. As shown in Figure 5, the white area on the surface of the samples further increased when the exposure time reached 12 months, and cracks appeared in the white area.

The cross-sectional morphologies of the samples are shown in Figure 6. When the exposure time reached 6 months, a relatively thick corrosion product layer formed on the surface of the samples, but it was relatively loose and internally contained a lot of holes and cracks. In addition, the corrosion product layer was uneven, and the surface was undulated greatly. When the exposure time reached 12 months, the corrosion product layer had obvious bulging, and part of the area was thinned due to the shedding of the corrosion product.

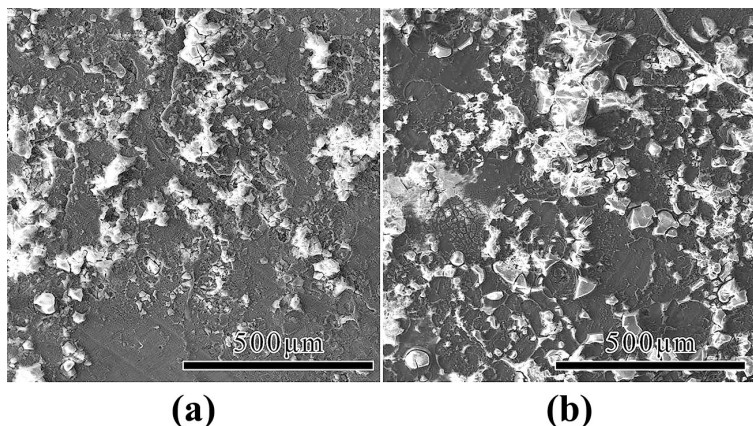

**Figure 5.** Surface morphologies of samples exposed for different long-term periods: (**a**) 6 months and (**b**) 12 months.

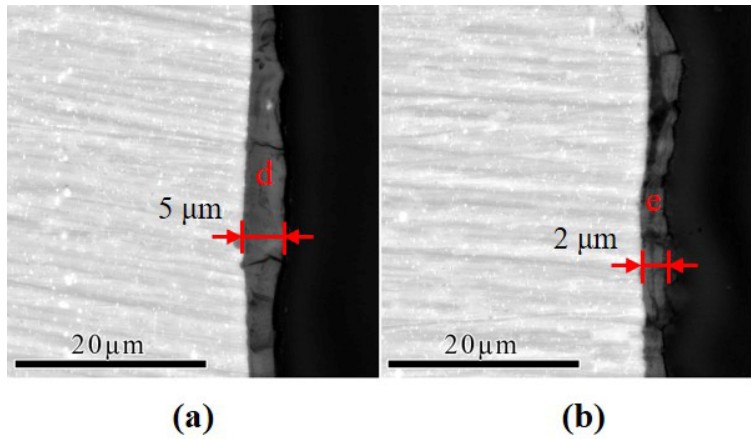

**Figure 6.** The cross-sectional morphologies of the samples exposed for different long-term periods: (**a**) 6 months and (**b**) 12 months.

### 3.3. Phase Analysis

Energy spectrum analysis was utilized on the red-lettered area in the microscopic images. Table 3 shows that, after 2 weeks of outdoor exposure, corrosion products appeared in the pits, but the amount was relatively small. The energy spectrum results contain the corrosion products and the sample matrix, and Table 3 shows that the weight percentage of oxygen was only 17.85%. When the exposure time reached 6 months, corrosion products accumulated on the surface of the samples to form a corrosion product layer; therefore, the weight percentage of oxygen increased to 70%. The phase analysis of the corrosion product powder on the surface of the samples exposed for 12 months in the outdoor environment was carried out. Figure 7 shows that the corrosion product has strong diffraction peaks generated by the matrix and $Al_2O_3 \cdot 3H_2O$. The resulting diffraction peaks indicate that the main component of the corrosion product is the monoclinic trihydrate $Al_2O_3 \cdot 3H_2O$. The results are consistent with previous studies in the field [23].

**Table 3.** Percentage of elements in the scanning area (wt.%).

| Area | O | Al | Mg | Cu | Mn |
|------|-------|-------|------|------|------|
| a | 17.85 | 78.18 | 1.52 | 2.23 | 0.23 |
| b | 71.56 | 25.04 | 1.2 | 0.8 | 0.12 |
| c | 71.77 | 26.5 | 1.31 | 0.3 | 0.12 |
| d | 70.85 | 25.13 | 2.61 | 0.95 | 0.38 |
| e | 70.72 | 25.2 | 1.95 | 1.84 | 0.29 |

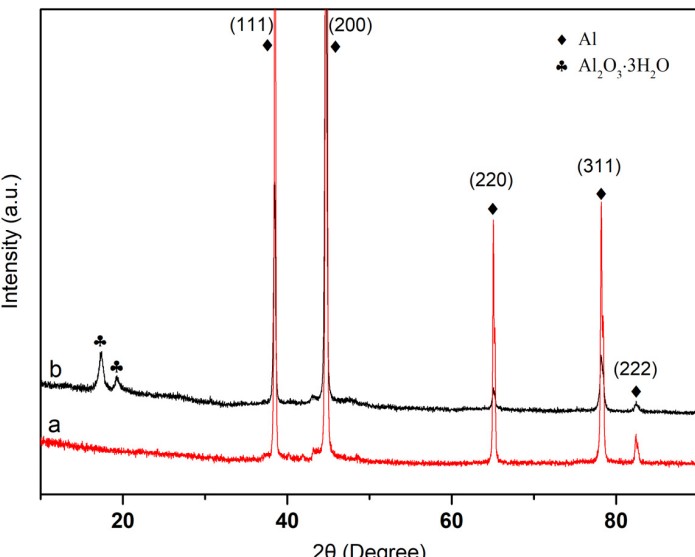

**Figure 7.** XRD test results of corrosion products of samples after different exposure times: 6 months (**a**) and 12 months (**b**).

### 3.4. EIS Analysis

The AC impedance spectra of samples exposed to different lengths of time in an outdoor environment are shown in Figure 8. The larger the impedance modulus value, the better the corrosion resistance. The figure shows that the initial samples have an inductive arc in the low-frequency region, indicating that the 2024 aluminum alloy is sensitive to Cl⁻ and that it entered a small pore corrosion induction period during the test. The initial samples did not undergo pitting; therefore, the capacitive reactance arc radius was the largest, and the impedance modulus value was the largest. After exposure for 1 month, the surface of the samples was pitted due to the erosion of Cl⁻; therefore, the capacitive reactance arc radius became smaller, and the impedance modulus became smaller. After exposure for three months, the capacitive reactance arc radius decreased, and the impedance modulus value decreased. When the exposure time reached 6 months, the surface of the samples was covered by corrosion products generated by pitting corrosion. This had a certain hindering effect on the diffusion of Cl⁻ into the interior; therefore, the capacitive reactance arc radius increased and the impedance modulus value increased. The increase in radius indicates an increase in corrosion resistance against a corrosive medium because the thickness of the corrosion product layer increased. As a result, the path of inward diffusion of Cl⁻ became longer. When the exposure time reached 12 months, the corrosion product layer began to fall off and thin, which weakened the barrier to Cl⁻ function. Hence, the capacitive reactance arc radius decreased again, and the impedance modulus value decreased.

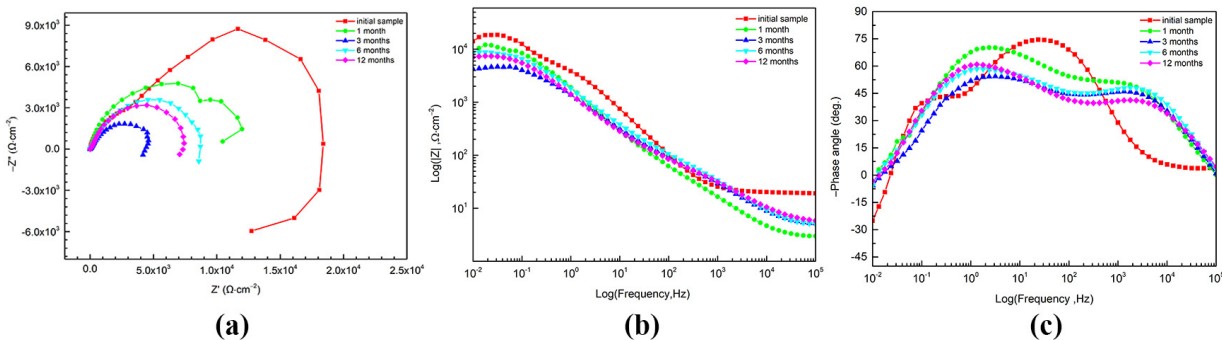

**Figure 8.** The AC impedance spectra of samples exposed to different lengths of time outdoors: (**a**) capacitive reactance arc, (**b**) impedance modulus value, and (**c**) phase angle.

Electrochemical analysis software was used to process the data. Based on the sensitivity of aluminum alloys to $Cl^-$, the equivalent circuit used for the initial samples is shown in Figure 9, where Rs is the solution resistance, Q is the constant phase angle element corresponding to the electric double layer, $R_1$ is the resistance of the charge passing through the surface oxide film, L is the equivalent inductance, and $R_2$ is the corresponding equivalent resistance [24–28]. Corresponding data are shown in Table 4.

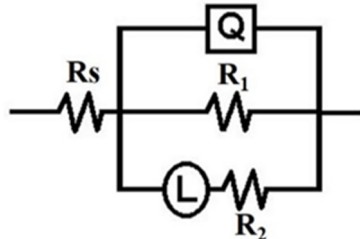

**Figure 9.** Equivalent circuit diagram for the initial state.

**Table 4.** Electrochemical parameters in the equivalent circuit diagram.

| Samples | $R_s$ $(\Omega \cdot cm^2)$ | Q $(\times 10^6 \, \Omega^{-1} \cdot cm^{-2})$ | n | $R_1$ $(k\Omega \cdot cm^2)$ | L $(H \cdot cm^2)$ | $R_2$ $(k\Omega \cdot cm^2)$ |
|---|---|---|---|---|---|---|
| Initial samples | 18.72 | 52.1 | 0.81 | 18.36 | $6.175 \times 10^5$ | 2.1 |

For one month after exposure, a pit corrosion structure occurred on the surface of the samples. The equivalent circuit used for the impedance spectrum of the samples is shown in Figure 10, where Rs is the solution resistance, $Q_1$ is the constant phase angle element corresponding to the electric double layer, $R_1$ is the resistance corresponding to the oxide film, $R_2$ is the solution resistance in the pitting pit, $Q_2$ is the constant phase angle element corresponding to the non-Faraday process in the pitting pit, and $R_3$ is the charge transfer resistance when the metal anode at the bottom of the pitting pit is dissolved. The specific values of each equivalent element are shown in Table 5.

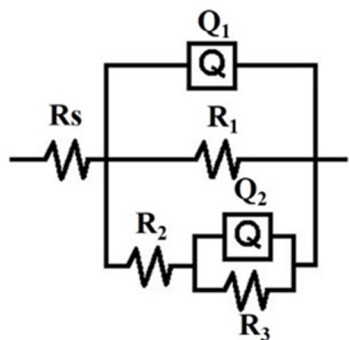

**Figure 10.** Equivalent circuit diagram for the specimens after three months of exposure.

**Table 5.** Electrochemical parameters in the equivalent circuit diagram.

| Samples | $R_s$ $(\Omega \cdot cm^2)$ | $Q_1$ $(\times 10^6 \, \Omega^{-1} \cdot cm^{-2})$ | $n_1$ | $R_1$ $(k\Omega \cdot cm^2)$ | $R_2$ | $Q_2$ $(k\Omega \cdot cm^2)$ | $n_2$ | $R_3$ $(k\Omega \cdot cm^2)$ |
|---|---|---|---|---|---|---|---|---|
| 1 month | 6.97 | 15.36 | 0.55 | 9.51 | 2.35 | 132.7 | 0.73 | 0.6 |
| 3 months | 7.93 | 4.716 | 0.84 | 4.58 | 3.74 | 60.61 | 0.79 | 1.91 |

When exposed to the outdoor environment for 6 months, the samples were generally corroded and the surface was covered with a corrosion product layer. Therefore, the equivalent circuit adopted is shown in Figure 11, where Rs is the solution resistance and

$Q_1$ is the normal corrosion product layer. For the phase angle element, $R_1$ is the transfer resistance of the charge in the corrosion product, $Q_2$ is the constant phase angle element corresponding to the electric double layer, and $R_2$ is the resistance corresponding to the oxide film [29–31]. Corresponding data are shown in Table 6.

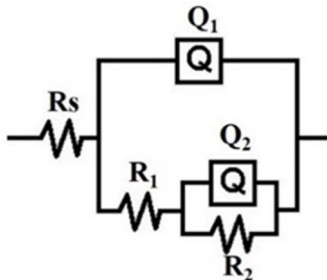

**Figure 11.** Equivalent circuit diagram for the specimens after six months of exposure.

**Table 6.** Electrochemical parameters in the equivalent circuit diagram.

| Samples | $R_s$ ($\Omega \cdot cm^2$) | $Q_1$ ($\times 10^6\ \Omega^{-1} \cdot cm^{-2}$) | $n_1$ | $R_1$ ($k\Omega \cdot cm^2$) | $Q_2$ ($k\Omega \cdot cm^{-2}$) | $n_2$ | $R_2$ ($k\Omega \cdot cm^2$) |
|---|---|---|---|---|---|---|---|
| 6 months | 4.64 | 38.19 | 0.75 | 12.51 | 90.73 | 0.75 | 5.23 |

The polarization curves of samples exposed to different lengths of time in an outdoor environment are shown in Figure 12, and the corresponding electrochemical parameters are shown in Table 7. In addition, according to the relevant equation, the corrosion rate was calculated in the table [32]. The self-corrosion current density is an important indicator for evaluating the corrosion resistance of the samples. The smaller the self-corrosion current density, the better the corrosion resistance of the samples. When the self-corrosion current density is the same, the higher the self-corrosion potential, the better the corrosion resistance of the samples. The figure shows that the initial samples had no pitting corrosion; therefore, the self-corrosion current density was the smallest. After one month of exposure, the surface of the samples had pitting corrosion and pitting pits; therefore, the self-corrosion current density of the samples began to increase. After 3 months of exposure, the number of pits increased, and the diameter of the pits further expanded. Therefore, the self-corrosion current density continued to increase. When the exposure time reached 6 months, the surface of the samples was covered by corrosion products generated by pitting corrosion. The corrosion product layer had a hindering effect on the inward diffusion of $Cl^-$ and at the same time, hindered the ion exchange process between the electrochemical corrosion reaction area and the electrolyte liquid membrane; therefore, the self-corrosion current density became smaller and the exposure time was longer. When it reached 12 months, the corrosion product layer peeled off and thinned due to internal stress; therefore, the hindering effect on the inward diffusion of $Cl^-$ and ion exchange began to weaken, and the self-corrosion current density increased again.

**Table 7.** Electrochemical parameters corresponding to the polarization curve.

| Samples | $E_{corr}$ (mV vs. SCE) | $I_{corr}$ ($\mu A/cm^2$) | Corrosion Rate ($\mu m/year$) |
|---|---|---|---|
| Initial samples | −629 | 0.456 | 4.97 |
| 1 month | −592 | 1.387 | 15.12 |
| 3 months | −593 | 8.338 | 90.88 |
| 6 months | −619 | 2.359 | 25.71 |
| 12 months | −555 | 2.849 | 31.05 |

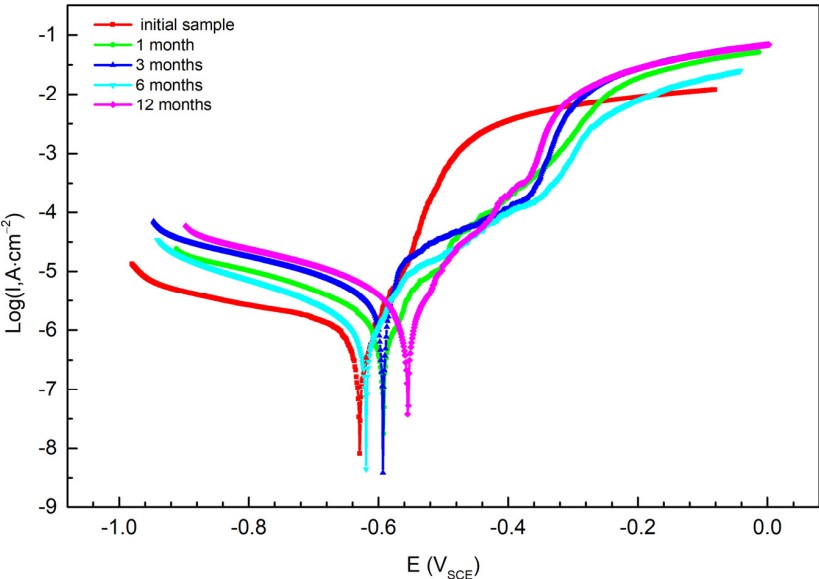

**Figure 12.** The polarization curves of samples exposed to different lengths of time in an outdoor environment.

*3.5. Corrosion Mechanism*

3.5.1. The Initiation of Pitting Corrosion

When the samples are placed in an outdoor environment, the humidity in the air in the marine atmosphere is relatively high, the air is filled with small droplets, and the droplets contain corrosive anions (mainly $Cl^-$). The droplets condense on the surface of the samples, $Cl^-$ is adsorbed to the surface of the oxide film, and the following chemical reaction occurs, which destroys and dissolves the oxide film on the surface of the samples. In this process, as the oxide film dissolves and becomes thinner, the impedance modulus decreases, the self-corrosion current density increases, and the corrosion resistance deteriorates.

$$Al(OH)_3 + Cl^- \rightarrow Al(OH)_2Cl + OH^- \tag{3}$$

$$Al(OH)_2Cl + Cl^- \rightarrow Al(OH)Cl_2 + OH^- \tag{4}$$

$$Al(OH)Cl_2 + Cl^- \rightarrow AlCl_3 + OH^- \tag{5}$$

After the oxide film is dissolved, the substrate of the samples leaks out. Since the equilibrium potential is lower than the adjacent oxide film, the oxidation reaction of the substrate as the anode is shown in Formula (7). The substrate continuously dissolves and forms pitting pits. The reaction continues to occur, the diameter of the pitting pits becomes larger, the number of pits increases, and the protective effect of the oxide film continues to weaken. This stage is manifested by the disappearance of the low-frequency inductive arc and the reduction of the impedance modulus, and the self-corrosion current density becomes larger, indicating that the corrosion resistance of the samples continues to deteriorate.

$$Al - 3e^- \rightarrow Al^{3+} \tag{6}$$

$$O_2 + 2H_2O + 4e^- \rightarrow 4OH^- \tag{7}$$

Rainfall wets the surface of the samples, which is conducive to the formation of an electrolyte liquid film on the surface of the samples, and creates conditions for the occurrence of corrosion. Therefore, pitting corrosion occurs quickly on the surface. However, sunlight exposure can also accelerate the evaporation of water in the electrolyte film on the surface of the samples since electrochemical corrosion can only occur on the electrolyte film. When the electrolyte liquid film decreases due to the accelerated evaporation of water, the occurrence of electrochemical corrosion is hindered; therefore, electrochemical corrosion cannot continue to occur. The pitting pits cannot continue to expand into the samples,

and it is difficult for intergranular corrosion to occur on the surface. As the exposure time increases, the surface of the samples begins to enter a general corrosion stage. Therefore, the electrolyte solution on the samples' surface contains a large amount of $Al^{3+}$ and OH ions; the reaction between the two ions is shown in the Formula (8). The product $Al(OH)_3$ will continue to react due to its instability, forming $Al_2O_3 \cdot 3H_2O$, which will accumulate on the surface of the samples to form a corrosion product layer, blocking the inward diffusion of $Cl^-$ and the ion exchange between the reaction area and the liquid membrane. Therefore, the impedance modulus of the samples increases at this stage, the self-corrosion current density decreases, and the corrosion resistance of the samples becomes better.

### 3.5.2. Spalling of the Corrosion Products

After a period of accumulation, the corrosion product layer gradually thickens, and the internal stress continues to increase. Finally, the corrosion product layer falls off and thins, which leads to a weakening of the inward diffusion of $Cl^-$ and ion exchange in the reaction zone. Therefore, the impedance modulus decreases, the self-corrosion current density increases, and the corrosion resistance of the samples deteriorates again.

$$Al^{3+} + 3OH^- \rightarrow Al(OH)_3 \tag{8}$$

$$2Al(OH)_3 \rightarrow Al_2O_3 \cdot 3H_2O \tag{9}$$

### 4. Conclusions

Based on the above results, the following conclusions can be drawn:

The corrosion product weight gain and exposure time of the 2024 aluminum alloy follow a power-law function. In the early stage, the aluminum alloy undergoes pit corrosion, generating some corrosion products, which then converge to form a loose corrosion product layer. Then, pit corrosion transforms into general corrosion, and electrochemical corrosion produces a large amount of $Al(OH)_3$, leading to the thickening of the loose corrosion product layer. In the middle and later stages of corrosion, the outer layer of corrosion products peeled off, leaving a dense stacked layer of corrosion products. The electrochemical test results show that, as the exposure time increased, the capacitance arc first contracts, then expands, and finally, contracts slightly. Correspondingly, the corrosion current density of the 2024 aluminum alloy first decreased, then increased, and finally, decreased again.

**Author Contributions:** Conceptualization, J.Z., T.Z. and Z.Z.; Methodology, T.Z., J.Z. and J.W.; Investigation, T.Z., J.Z., Y.Z. and J.W.; Data curation, T.Z. and J.Z.; Writing—original draft preparation, J.Z. and J.W.; Writing—review and editing, J.Z., Z.C. and M.C.; Visualization, J.W. and M.C. All authors have read and agreed to the published version of the manuscript.

**Funding:** This project was financially supported by the National Natural Science Foundation of China under Grant (51671053 and 51801021), the Fundamental Research Funds for the Central Universities (No. N2302007), and the Ministry of Industry and Information Technology Project (No. MJ-2017-J-99).

**Institutional Review Board Statement:** Not applicable.

**Informed Consent Statement:** Not applicable.

**Data Availability Statement:** Data are contained within the article.

**Conflicts of Interest:** The authors declare that they have no known competing financial interests or personal relationships that could have appeared to influence the work reported in this paper.

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
