# Peer review of "Corrosion Behavior of the 2024 Aluminum Alloy in the Atmospheric Environment of the South China Sea Islands"

_coatings, doi:10.3390/coatings14030331_

Round 1
Reviewer 1 Report
Comments and Suggestions for Authors
The manuscript is of no interest from either a scientific or practical point of view. The mechanism of pitting corrosion of aluminum alloys in chloride solutions or in a coastal atmosphere has been known for many years; it is described in dozens of monographs and textbooks. It should be emphasized that the authors of this manuscript present this mechanism at a very primitive level.
For solving engineering problems, for example, predicting the corrosion rate of a 2024 alloy in the atmosphere, the results of this study are also not of interest. The corrosion rate of the alloy in the atmosphere has not been determined. The increase in the masses of the samples does not necessarily have to be proportional to the corrosion rate, since some of the easily soluble corrosion products are washed away by rain. The corrosion inhibition coefficient (power exponent n value) determined by the authors is also not of interest, since for long-term prediction of the corrosion resistance of a material it is necessary to determine the coefficient n over several years, until a stationary corrosion rate is established.
Author Response
We sincerely thank you for your letter and the reviewers’ valuable comments and suggestions to improve the quality of our manuscript. Here we submit our manuscript with the title "Corrosion Behavior of 2024 Aluminum Alloy in the Atmosphere Environment of South China Sea Islands". We have made serious and extensive corrections based on the comments and hope to satisfy your request. The main revisions are highlighted in red color in the revised manuscript and point-by-point responses to the comments are listed as follows:
Before answering all the questions, we would like to express my gratitude to the editor and all the reviewers for giving us the opportunity to fully revise the paper and all the feedback. I have carefully read all the comments and the questions raised by the four reviewers are very profound and professional. Here I would like to explain the original intention of writing this article. 2024 aluminum alloy is a high-strength aluminum alloy that can be used to manufacture aircraft fuselage, but due to the presence of strengthening phases, its corrosion resistance is poor. In recent years, the corrosion of metal materials used on airplanes has been a hot research direction in corrosion field. The South China Sea is an area with extremely harsh corrosive environments, due to the high annual average temperature and high concentration of Cl-. However, there are few reports on its corrosion data in literatures. For this reason, this article used conventional corrosion behavior research methods to study it, including kinetic curves, microstructure, phase analysis, and electrochemical testing. As the reviewer pointed out, the duration of studying atmospheric corrosion rule should be at least in years in order to obtain stationary data. However, this experiment was just conducted, so only preliminary early corrosion data has been obtained. In order to investigate the early corrosion rule as soon as possible, this article was completed. As the experimental time increases, corrosion data will be further accumulated. We will continue to improve our experimental results and provide a complete study of corrosion rule of 2024 aluminum alloy. I know there are still many shortcomings in this article, so the article has been carefully revised according to the reviewers' comments and suggestions. Next, I will continue to make every effort to improve it. Thank you for your valuable feedback and continued attention.
Reviewer #1:
Major issue 1: The manuscript is of no interest from either a scientific or practical point of view. The mechanism of pitting corrosion of aluminum alloys in chloride solutions or in a coastal atmosphere has been known for many years; it is described in dozens of monographs and textbooks. It should be emphasized that the authors of this manuscript present this mechanism at a very primitive level.
Response 1: Thank you for your comments. Here, I would like to explain the original intention of writing this article. 2024 aluminum alloy is a high-strength aluminum alloy that can be used to manufacture aircraft fuselage, but due to the presence of strengthening phases, its corrosion resistance is poor. In recent years, the corrosion of metal materials used on airplanes has been a hot research direction in corrosion field. The South China Sea is an area with extremely harsh corrosive environments, due to the high annual average temperature and high concentration of Cl-. However, there are few reports on its corrosion data in literatures. For this reason, this article used conventional corrosion behavior research methods to study it, including kinetic curves, microstructure, phase analysis, and electrochemical testing.
Major issue 2: For solving engineering problems, for example, predicting the corrosion rate of a 2024 alloy in the atmosphere, the results of this study are also not of interest. The corrosion rate of the alloy in the atmosphere has not been determined. The increase in the masses of the samples does not necessarily have to be proportional to the corrosion rate, since some of the easily soluble corrosion products are washed away by rain. The corrosion inhibition coefficient (power exponent n value) determined by the authors is also not of interest, since for long-term prediction of the corrosion resistance of a material it is necessary to determine the coefficient n over several years, until a stationary corrosion rate is established.
Response 2: Thank you for your comments. As you pointed out, the duration of studying atmospheric corrosion rule should be at least in years in order to obtain stationary data. However, this experiment was just conducted, so only preliminary early corrosion data has been obtained. In order to investigate the early corrosion rule as soon as possible, this article was completed. As the experimental time increases, corrosion data will be further accumulated. We will continue to improve our experimental results and provide a complete study of corrosion rule of 2024 aluminum alloy.

Reviewer 2 Report
Comments and Suggestions for Authors
attached

Revisiting grammar and writing format is advised, as there are numerous grammatical errors throughout the document.
Author Response
Dear Editor and Reviewers:
We sincerely thank you for your letter and the reviewers’ valuable comments and suggestions to improve the quality of our manuscript. Here we submit our manuscript with the title "Corrosion Behavior of 2024 Aluminum Alloy in the Atmosphere Environment of South China Sea Islands". We have made serious and extensive corrections based on the comments and hope to satisfy your request. The main revisions are highlighted in red color in the revised manuscript and point-by-point responses to the comments are listed as follows:
Before answering all the questions, we would like to express my gratitude to the editor and all the reviewers for giving us the opportunity to fully revise the paper and all the feedback. I have carefully read all the comments and the questions raised by the four reviewers are very profound and professional. Here I would like to explain the original intention of writing this article. 2024 aluminum alloy is a high-strength aluminum alloy that can be used to manufacture aircraft fuselage, but due to the presence of strengthening phases, its corrosion resistance is poor. In recent years, the corrosion of metal materials used on airplanes has been a hot research direction in corrosion field. The South China Sea is an area with extremely harsh corrosive environments, due to the high annual average temperature and high concentration of Cl-. However, there are few reports on its corrosion data in literatures. For this reason, this article used conventional corrosion behavior research methods to study it, including kinetic curves, microstructure, phase analysis, and electrochemical testing. As the reviewer pointed out, the duration of studying atmospheric corrosion rule should be at least in years in order to obtain stationary data. However, this experiment was just conducted, so only preliminary early corrosion data has been obtained. In order to investigate the early corrosion rule as soon as possible, this article was completed. As the experimental time increases, corrosion data will be further accumulated. We will continue to improve our experimental results and provide a complete study of corrosion rule of 2024 aluminum alloy. I know there are still many shortcomings in this article, so the article has been carefully revised according to the reviewers' comments and suggestions. Next, I will continue to make every effort to improve it. Thank you for your valuable feedback and continued attention.
Reviewer #2:
Major issue 1: Lines 63 and 64: The author stated, "There are few reports on the corrosion behavior of 2024 aluminum alloy in the South China Sea atmosphere using the field exposure method." The author didn't mention the novelty of the work clearly; please state how this work differs from the previous reports and the significance of these results. Also,please add the earlier reports as references.
Response 1: Thank you for your suggestion. 2024 aluminum alloy is a high-strength aluminum alloy that can be used to manufacture aircraft fuselage, but due to the presence of strengthening phases, its corrosion resistance is poor. In recent years, the corrosion of metal materials used on airplanes has been a hot research direction in corrosion field. The South China Sea is an area with extremely harsh corrosive environments, due to the high annual average temperature and high concentration of Cl-. However, there are few reports on its corrosion data in literatures. For this reason, this article used conventional corrosion behavior research methods to study it, including kinetic curves, microstructure, phase analysis, and electrochemical testing.
Major issue 2: The introduction does not clarify why this specific field's (south China) exposure method was investigated. The author may consider adding a comment on that in the introduction section.
Response 2: Thank you for your suggestion. The added content has been highlighted in red in the article.
Major issue 3: Line 125: Weight gain per unit area increases with time prolongation. There is no clear explanation for this behavior. The author may consider adding a comment on this.
Response 3: Thank you for your suggestion. The increase in mass is due to the accumulation of corrosion products on the surface of the sample, the added content has been highlighted in red in the article.
Major issue 4: The authors repeat the same thing in the "3.2 Microstructure" section. No technical explanation is provided in microscopic images and SEM images.
Response 4: Thank you for your comments. In the "3.2 Microstructure" section, the macroscopic morphology of the sample, as well as the microscopic morphology of the sample surface and cross-section are provided, it can be seen that pitting corrosion has occurred, and the size of the pits is approximately measured.
Major issue 5: In Figure 3, the captions are not proper. There are no particulars about the image's representation. The author stated in line 163, "where the red letters in the figure are the scanning area of the energy spectrum," but there are no red marks in the figure. Also, no energy spectrum is presented in this manuscript.
Response 5: Thank you for your comments. The specific information about the image has been supplemented in the caption. The red mark has been inserted into the image, and the energy spectrum information is shown in Table 3.
Major issue 6: Line 195 to 197: "Due to this morphology, Al3+ and OH- newly formed by the corrosion electrochemical reaction can penetrate the gaps of the corrosion products to form new corrosion products." How did the author confirm that Al3+ and OH- are newly formed in the SEM images?
Response 6: Thank you for your comments. Due to the increasing amount of corrosion products, it can be inferred that the electrochemical corrosion reaction is constantly occurring. If the new corrosion product is not formed inside the corrosion product layer, bulges or cracks will not occur.
Major issue 7: In line 213, the author again mentioned, "marked in a red letter." But there are no red marks in any images. It is hard to understand without proper representation.
Response 7: Thank you for your comments. The red mark has been inserted into the image, and the energy spectrum information is shown in Table 3.
Major issue 8: In XRD analysis, the author presented the XRD pattern for only one sample (12 months of exposure), but no reference pattern exists. It is better to compare with other samples that were studied in this work, such as 2024 aluminum, one week, two weeks, one month, and six months.
Response 8: Thank you for your suggestion. In fact, corrosion product layer formed on the sample is very thin, the XRD result only shows aluminum. To be characterized more precisely, we scrape the corrosion products off the surface of the sample. The XRD results in the article are based on the XRD analysis of corrosion products. As for one week, two weeks,one month, and six months’ samples, corrosion products are few, which are not sufficient for XRD testing.
Major issue 9: In line 240, the reason for the increase in reactance arc radius for six months of exposure is not explained in detail. In line 291, The author states, "The corrosion product layer has a certain hindering effect on the inward diffusion of Cl-" "Add references and provide a clearer explanation on that "certain hindering effect on the inward diffusion of Cl-."
Response 9: Thank you for your suggestion. The increase in radius indicates an increase in corrosion resistance against corrosive medium, which is because the thickness of the corrosion product layer increases. As a result, the path of inward diffusion of Cl- becomes longer. The explanation and has been added and highlighted in red in the article.
Other suggestions
In Figure 4, please mention in the caption what the exposure time is for (a), (b), (c), and (d).
Response: The article has been revised as the suggestion.
In line 129: Add figure number.
Response: The article has been revised as the suggestion.
Line 221: The figure number is supposed to be 7, not 3.7.
Response: The article has been revised as the suggestion.
In lines 231 to 233: Please add references.
Response: The structure of the article has changed due to the addition of content. So lines 231 to 233 were copied in here. It is “It can be seen from the figure that the initial sample has an inductive arc in the low-frequency region, indicating that the 2024 aluminum alloy is sensitive to Cl- ions, and it has entered a small pore corrosion induction period during the test.” The explanation is from a book named Principles of Electrochemistry of Corrosion written by Cao Chunan.
In lines 280-281: Please add references.
Response: Lines 280-281 is “The smaller the self-corrosion current density, the better the corrosion resistance of the sample.” This is also from the book as above.
Lines 308 to 310: Add references, please.
Response: In this process, as the oxide film dissolves and becomes thinner, the impedance modulus decreases, the self-corrosion current density increases, and the corrosion resistance deteriorates. This sentence is the same statement as the above sentence.
Line 313: The formula number is incorrect.
Response: The article has been revised as the suggestion.
Line 332: The formula number is incorrect.
Response: The article has been revised as the suggestion.
Line 333: "Forming Al2O3 with crystal water." what does the author mean by crystal water?
Response: It is Al2O3·3H2O, the article has been revised as the suggestion.
Line 332 to 337: Please add references.
Response: Lines 332-337 is “The product Al(OH)3 will continue to react due to its instability, forming Al2O3 with crystal water, which will accumulate on the surface of the sample to form a corrosion product layer, blocking the inward diffusion of Cl- and the ion exchange between the reaction area and the liquid membrane, so the impedance modulus of the sample increases at this stage, the self-corrosion current density decreases, and the corrosion resistance of the sample becomes better.” In fact, the blocking effect of the corrosion product layer on Cl- is widely recognized.
Line 339 to 341: Please add references.
Response: Lines 339-341 is “Because the corrosion product layer is relatively loose and contains a large number of holes and crevices inside, the corrosion medium containing Al3+ and OH- ions will penetrate into the pores and crevices in the corrosion product layer to generate new corrosion products.” Al3+ and OH- are soluble in water, and water can diffuse inward in the corrosion product layer, so Al3+ and OH- are brought into the interior of the corrosion layer.
Line 349: It is "4. Corrosions" Or "conclusions"
Response: Conclusions, the article has been revised as the suggestion.

Reviewer 3 Report
Comments and Suggestions for Authors
This paper was well prepared and organized from the point that it showed the atmospheric corrosion of AA2024. But some part of the manuscript needs to be revised as follows.
1. Abstract; 'Uniform pitting' may have a misunderstading because pitting corrosion is a representative localized corrosion. Please rewrite it.
2.Materials;
1) How did you refine the surface?
2) Table 1 needs the analyzed chemical composition.
3) Table 2. How is the corrosivity in this area according to ISO standard?
4) 2.2 Corrosion weight loss test is not correct. Please use 'Atmospheric corrosion test'.
3. Methods
1) All instruments used need the detail information about 'model, manufacturer, country'.
2) 2.4 X-ray diffraction; What is the scan rate? 0.02 degree/min or not?
3) 2.5.1 AC impedance; Did you deaerate the test solution or not? If not, why?
4) 2.5.2 Polarization test, not curve test (may be not Tafel test). Did you deaerate test solution? What are counter electrode and reference electrode? How did you treat the surface before electrochemical test? (immersion for 1hour or cathodic polarized?) What is test solution? Why did you use that solution?
4. Results
1) 3.2 is Surface morphologies, NOT 'Microstructure'.
2) Figure 2 is a little small.
3) Figure 3, Figure 4. What are (a)(b)(c)(d)?
4) Figure 4 and FIgure 6 revealed the result for different specimen? or Same results?
5) Figure 5 and Table 3; Did you get SEM-EDS on the surface corrosion products? Where are a,b,c,d,e area?
6) FIgure 7; Because 1 specimen was plotted, correct the unit form a.u. to c.p.s.
7) Why did you use the models like Figure 9 and Figure 10 and Figure 11? Please explain why the equivalent model was changed with exposure time and refer the related references.
8) Why did you get the data in Table 7? Was the corrosion rate in FIgure 1 same to those in Table 7? If not, explain the reason.
9) Check the supercript in the whole manuscript.
5. Conclusions is not the summary of the results. Based on the above points, the conclusions need to be deeper and clearer on also the drawbacks of this process, it may be fine to generally promote this process, but the authors should provide also a comprehensive and objective list of conclusions with the good the bad and the neutral conclusions.
6. References are not sufficient. Please survey the related research papers by other country.
Author Response
Dear Editor and Reviewers:
We sincerely thank you for your letter and the reviewers’ valuable comments and suggestions to improve the quality of our manuscript. Here we submit our manuscript with the title "Corrosion Behavior of 2024 Aluminum Alloy in the Atmosphere Environment of South China Sea Islands". We have made serious and extensive corrections based on the comments and hope to satisfy your request. The main revisions are highlighted in red color in the revised manuscript and point-by-point responses to the comments are listed as follows:
Before answering all the questions, we would like to express my gratitude to the editor and all the reviewers for giving us the opportunity to fully revise the paper and all the feedback. I have carefully read all the comments and the questions raised by the four reviewers are very profound and professional. Here I would like to explain the original intention of writing this article. 2024 aluminum alloy is a high-strength aluminum alloy that can be used to manufacture aircraft fuselage, but due to the presence of strengthening phases, its corrosion resistance is poor. In recent years, the corrosion of metal materials used on airplanes has been a hot research direction in corrosion field. The South China Sea is an area with extremely harsh corrosive environments, due to the high annual average temperature and high concentration of Cl-. However, there are few reports on its corrosion data in literatures. For this reason, this article used conventional corrosion behavior research methods to study it, including kinetic curves, microstructure, phase analysis, and electrochemical testing. As the reviewer pointed out, the duration of studying atmospheric corrosion rule should be at least in years in order to obtain stationary data. However, this experiment was just conducted, so only preliminary early corrosion data has been obtained. In order to investigate the early corrosion rule as soon as possible, this article was completed. As the experimental time increases, corrosion data will be further accumulated. We will continue to improve our experimental results and provide a complete study of corrosion rule of 2024 aluminum alloy. I know there are still many shortcomings in this article, so the article has been carefully revised according to the reviewers' comments and suggestions. Next, I will continue to make every effort to improve it. Thank you for your valuable feedback and continued attention.
Reviewer #3:
- Abstract; 'Uniform pitting' may have a misunderstanding because pitting corrosion is a representative localized corrosion. Please rewrite it.
Response: Thank you for your suggestion. It has been revised.
2.Materials;
1) How did you refine the surface?
Response: The surface roughness of the sample was provided by the manufacturer.
2) Table 1 needs the analyzed chemical composition.
Response: Thank you for your suggestion. As the presence of second phases in 2024 aluminum alloys, energy spectrum analysis is limited and nominal chemical composition can more directly present the elemental compositions.
3) Table 2. How is the corrosivity in this area according to ISO standard?
Response: Extreme, CX, according to ISO9223-2012.
4) 2.2 Corrosion weight loss test is not correct. Please use 'Atmospheric corrosion test'.
Response: Thank you for your suggestion. It has been revised.
- Methods
1) All instruments used need the detail information about 'model, manufacturer, country'.
Response: Thank you for your suggestion. It has been revised in the article.
2) 2.4 X-ray diffraction; What is the scan rate? 0.02 degree/min or not?
Response: Thank you for your suggestion. The scan rate is 8°/min and a step of 0.02°/point. It has been revised in the article.
3) 2.5.1 AC impedance; Did you deaerate the test solution or not? If not, why?
Response: Not. The deionized water used for preparing the solution is fresh and the oxygen content is small, which can the not affect testing.
4) 2.5.2 Polarization test, not curve test (may be not Tafel test). Did you deaerate test solution? What are counter electrode and reference electrode? How did you treat the surface before electrochemical test? (immersion for 1hour or cathodic polarized?) What is test solution? Why did you use that solution?
Response: Thank you for your comments. The test solution wasn’t deaerated. The counter electrode is a platinum electrode, and the reference electrode is a saturated calomel electrode. Before conducting the electrochemical test, the open circuit potential was measured and the test began when the open circuit potential remained stable for 30 minutes.Test solution is 3.5 wt.% NaCl solution, because it is close to the composition of seawater.
- Results
1) 3.2 is Surface morphologies, NOT 'Microstructure'.
Response: Thank you for your suggestion. It has been revised.
2) Figure 2 is a little small.
Response: Thank you for your suggestion. It has been revised.
3) Figure 3, Figure 4. What are (a)(b)(c)(d)?
Response: Thank you for your suggestion. The information has been added in the caption.
4) Figure 4 and Figure 6 revealed the result for different specimen? or Same results?
Response: Figure 4 presents samples of 1 week, 2 weeks, 3 weeks and 1 month. Figure 6 presents samples of 6 months and 12 months.
5) Figure 5 and Table 3; Did you get SEM-EDS on the surface corrosion products? Where are a,b,c,d,e area?
Response: The information has been added in the Figure 3, Figure 4 and Figure 6.
6) Figure 7; Because 1 specimen was plotted, correct the unit form a.u. to c.p.s.
Response: Thank you for your suggestion. It has been revised.
7) Why did you use the models like Figure 9 and Figure 10 and Figure 11? Please explain why the equivalent model was changed with exposure time and refer the related references.
Response: Thank you for your suggestion. As the corrosion product layer become thicker, the hindering effect on the inward diffusion of chloride ions will also change, so the equivalent circuit will change correspondingly.
8) Why did you get the data in Table 7? Was the corrosion rate in Figure 1 same to those in Table 7? If not, explain the reason.
Response: Thank you for your comments. The data in Table 7 can more intuitively display the results of the Tafel plots in Figure 12. The trend of corrosion rate reflected by the data in Table 7 is consistent with that in Figure 1. As the thickness of the corrosion product layer increased, the corrosion rate slowed down.
9) Check the supercript in the whole manuscript.
Response: Thank you for your suggestion. It has been revised.
- Conclusions is not the summary of the results. Based on the above points, the conclusions need to be deeper and clearer on also the drawbacks of this process, it may be fine to generally promote this process, but the authors should provide also a comprehensive and objective list of conclusions with the good the bad and the neutral conclusions.
Response: Thank you for your suggestion. It has been revised.
- References are not sufficient. Please survey the related research papers by other country.
Response: Thank you for your suggestion. It has been revised.

Reviewer 4 Report
Comments and Suggestions for Authors
The manuscript has the following reasons to reject.
1. The references are not sufficient. The authors should know that limited number of references (only 7 in this case) will not make a manuscript to be accepted.
2. Figure 2 is of very low quality. Phase analysis of the alloy prior to the corrosion required.
Comments on the Quality of English LanguageThe English writing skill should be improved.
Author Response
Dear Editor and Reviewers:
We sincerely thank you for your letter and the reviewers’ valuable comments and suggestions to improve the quality of our manuscript. Here we submit our manuscript with the title "Corrosion Behavior of 2024 Aluminum Alloy in the Atmosphere Environment of South China Sea Islands". We have made serious and extensive corrections based on the comments and hope to satisfy your request. The main revisions are highlighted in red color in the revised manuscript and point-by-point responses to the comments are listed as follows:
Before answering all the questions, we would like to express my gratitude to the editor and all the reviewers for giving us the opportunity to fully revise the paper and all the feedback. I have carefully read all the comments and the questions raised by the four reviewers are very profound and professional. Here I would like to explain the original intention of writing this article. 2024 aluminum alloy is a high-strength aluminum alloy that can be used to manufacture aircraft fuselage, but due to the presence of strengthening phases, its corrosion resistance is poor. In recent years, the corrosion of metal materials used on airplanes has been a hot research direction in corrosion field. The South China Sea is an area with extremely harsh corrosive environments, due to the high annual average temperature and high concentration of Cl-. However, there are few reports on its corrosion data in literatures. For this reason, this article used conventional corrosion behavior research methods to study it, including kinetic curves, microstructure, phase analysis, and electrochemical testing. As the reviewer pointed out, the duration of studying atmospheric corrosion rule should be at least in years in order to obtain stationary data. However, this experiment was just conducted, so only preliminary early corrosion data has been obtained. In order to investigate the early corrosion rule as soon as possible, this article was completed. As the experimental time increases, corrosion data will be further accumulated. We will continue to improve our experimental results and provide a complete study of corrosion rule of 2024 aluminum alloy. I know there are still many shortcomings in this article, so the article has been carefully revised according to the reviewers' comments and suggestions. Next, I will continue to make every effort to improve it. Thank you for your valuable feedback and continued attention.
Reviewer #4:
Major issue 1: The references are not sufficient. The authors should know that limited number of references (only 7 in this case) will not make a manuscript to be accepted.
Response 1: Thank you for your suggestion. It has been revised.
Major issue 2: Figure 2 is of very low quality. Phase analysis of the alloy prior to the corrosion required.
Response 2: Thank you for your suggestion. It has been revised.

Round 2
Reviewer 1 Report
Comments and Suggestions for Authors
Accept in present form
Author Response
Thank you for your comments. Please refer to the attachment for specific reply.
Reviewer 2 Report
Comments and Suggestions for Authors
The author failed to address all the inquiries raised by the reviewers in the previous round. Please do. The author should pay attention to each comment and make necessary changes in the manuscript. Here are a few comments that the author should add to the manuscript with proper explanation.
1. The author's explanation regarding the novelty of the work remains unclear, as noted in previous comments. While the author repeats the novelty statement, it lacks clarity on how this work differs from previous reports. It is crucial to incorporate the novelty of this work into the introduction section for better context and understanding.
2. The author has presented eight microscopic images in Figure 2. However, what each image stands for or what each picture represents is not at all clear. Why did the author remove the caption in all the images? The author should consider adding a label on each image for better understanding.
3. Line 201 to 203: "Due to this morphology, Al3+ and OH- newly formed by the corrosion electrochemical reaction can penetrate the gaps of the corrosion products to form new corrosion products." The author concluded without any proper experimental results. The author should provide some experimental analysis to prove the newly formed Al3+ and OH-.
4. In Table 3, the O wt% reaches 70.85 in d, corresponding to six months of exposure. Then, the O wt% did not increase further, and there is no explanation for this behavior. The author should add a comment on this.
5. The author should consider presenting XRD data for samples exposed for at least two different time durations (e.g., 6 months and 1 year) to investigate phase formation. While Figures 6a and 6b indicate the formation of a corrosion layer, XRD measurements for these samples would provide valuable insight. Additionally, including XRD data for samples exposed for 1 month, even with low corrosion product on the surface, would enhance understanding of phase formation.
6. The manuscript contains numerous typos. The author should thoroughly review the entire manuscript to manage these errors.
7. The author should define abbreviations when they are first used. For example, 'Al' was not defined in the introduction but was used in line 67.
Author Response
Reviewer #2:
The author failed to address all the inquiries raised by the reviewers in the previous round. Please do. The author should pay attention to each comment and make necessary changes in the manuscript. Here are a few comments that the author should add to the manuscript with proper explanation.
Response: Thank you for your professional suggestions. They have benefited me a lot and made this manuscript more accurate. The manuscript has undergone three revisions, and currently version is manuscript-R3. All the comments have been adopted. The modified parts have been highlighted in red.
- The author's explanation regarding the novelty of the work remains unclear, as noted in previous comments. While the author repeats the novelty statement, it lacks clarity on how this work differs from previous reports. It is crucial to incorporate the novelty of this work into the introduction section for better context and understanding.
Response: Thank you for your suggestion. The South China Sea has extremely harsh corrosive environments because the average temperature is 27 °C and the highest temperatures can be reached above 35 °C. The average value of relative humidity (RH) is 77%, and the highest humidity is 85%. Corrosivity of the environment is CX, according to ISO9223-2012. However, there are few reports on the corrosion behavior of 2024 aluminum alloy in the atmosphere of the South China Sea using on-site exposure methods. The above contents have been added in manuscript-R3.
- The author has presented eight microscopic images in Figure 2. However, what each image stands for or what each picture represents is not at all clear. Why did the author remove the caption in all the images? The author should consider adding a label on each image for better understanding.
Response: Thank you for your suggestion. The caption has been added in Figure 2.
- Line 201 to 203: "Due to this morphology, Al3+ and OH- newly formed by the corrosion electrochemical reaction can penetrate the gaps of the corrosion products to form new corrosion products." The author concluded without any proper experimental results. The author should provide some experimental analysis to prove the newly formed Al3+ and OH-.
Response: Thank you for your suggestion. This inference is indeed lack of evidence, so I deleted this sentence in manuscript-R3. However, as can be seen from Figure 6, there are many cracks in the corrosion product layer, which is its structural characteristics.
- In Table 3, the O wt% reaches 70.85 in d, corresponding to six months of exposure. Then, the O wt% did not increase further, and there is no explanation for this behavior. The author should add a comment on this.
Response: Thank you for your suggestion. In the manuscript-3, the EDS results and XRD jointly demonstrate the composition of corrosion products, and it is normal for the content of Oxygen to change within a certain range. In addition, the EDS results were compared with relevant literature, and the element content was basically similar. Here is the reference: doi.org/10.1016/j.jallcom.2019.03.028.
- The author should consider presenting XRD data for samples exposed for at least two different time durations (e.g., 6 months and 1 year) to investigate phase formation. While Figures 6a and 6b indicate the formation of a corrosion layer, XRD measurements for these samples would provide valuable insight. Additionally, including XRD data for samples exposed for 1 month, even with low corrosion product on the surface, would enhance understanding of phase formation.
Response: Thank you for your suggestion. XRD data of sample exposed for 6 months has been added in Figure 7.
- The manuscript contains numerous typos. The author should thoroughly review the entire manuscript to manage these errors.
Response: Thank you for your suggestion. I have carefully revised the grammar and vocabulary of the article again.
- The author should define abbreviations when they are first used. For example, 'Al' was not defined in the introduction but was used in line 67.
Response: Thank you for your suggestion. It has been revised in the manuscript-R3.
Reviewer 3 Report
Comments and Suggestions for Authors
1. Chemical composition of the test specimen MUST be analyzed using AAS etc, NOT spectromenter.
2. Corrosivity of tested area MUST be described in the manuscript, not to me only.
3. SEM (~~~ U.S.A.) in the manuscript
4. I don't understand your answer about the deaeration and small oxygen content because of deionization. This is not scientific answer. Please review the importance of deaeration in corrosion test. "The test solution was not deaerated" in the manuscript is more better even though the authors didn't realize the meaning and importance of deaeration. Please describe it in the manuscript.
5. 2.5.2 Polarization test is correct (curve test and Tafel test are wrong). The detail experimental process MUST be written in the manuscript, NOT to me. If you don't correct them, this manuscript may be rejected.
6. Impedance measurement; Please use one kind of model and explain the result. If the result didn't obey the model slightly, explain those phenomena. If each test uses the different models like that, a lot of equivalent cirvcuits may be needed.
7. Table 7; Ecorr, mV(SCE)
8. Did you revise the superscript? Really???
9. Conclusion is not the summary. Please conclude the generalized something.
10. Did you add the references? 12-references are still not sufficient.
Author Response
Reviewer #3:
Major issue 1: Chemical composition of the test specimen MUST be analyzed using AAS etc, NOT spectromenter.
Response 1: Thank you for your comments. Due to limitations in school equipment, the composition table adopted in the article is just provided by the manufacturer. To confirm accuracy, references 21-23 were supplemented to identify the composition intervals.
Major issue 2: Corrosivity of tested area MUST be described in the manuscript, not to me only.
Response 2: Thank you for your suggestion. The added content has been highlighted in red in line 66
Major issue 3: SEM (~~~ U.S.A.) in the manuscript
Response 3: Thank you for your suggestion. It has been revised.
Major issue 4: I don't understand your answer about the deaeration and small oxygen content because of deionization. This is not scientific answer. Please review the importance of deaeration in corrosion test. "The test solution was not deaerated" in the manuscript is more better even though the authors didn't realize the meaning and importance of deaeration. Please describe it in the manuscript.
Response 4: Thank you for your comments. Relevant literatures showed that deaeration can reduce the occurrence of oxygen uptake reactions. This does have an impact on the test results, therefore, "The test solution was not deaerated" has been added in the article,
Major issue 5: 2.5.2 Polarization test is correct (curve test and Tafel test are wrong). The detail experimental process MUST be written in the manuscript, NOT to me. If you don't correct them, this manuscript may be rejected.
Response 5: Thank you for your comments. It has been revised.
Major issue 6: Impedance measurement; Please use one kind of model and explain the result. If the result didn't obey the model slightly, explain those phenomena. If each test uses the different models like that, a lot of equivalent circuits may be needed.
Response 6: Thank you for your comments. The explanation and source literatures on the equivalent circuit model have been added to the article. The change in the circuit model is due to the changes in the structure of the corrosion products on the 2024 aluminum alloy. The thickness of corrosion products changed with the exposure time, so the equivalent circuit also changes.
Major issue 7: Table 7; Ecorr, mV(SCE)
Response 7: Thank you for your suggestion. It has been revised.
Major issue 8: Did you revise the superscript? Really???
Response 8: Thank you for your suggestion. I have carefully revised the grammar and vocabulary of the article again. The modified content has been highlighted in red in the article.
Major issue 9: Conclusion is not the summary. Please conclude the generalized something.
Response 9: Thank you for your suggestion. It has been revised.
Major issue 10:Did you add the references? 12-references are still not sufficient.
Response 10: Thank you for your suggestion. More references have been added to the article.

Reviewer 4 Report
Comments and Suggestions for Authors
The authors still need to improve the manuscript. Only 12 references are not upto the level of the journal, like "Coatings".
Comments on the Quality of English LanguageStill little modifcations are needed.
Author Response
Reviewer #4:
- Major issue 1: The authors still need to improve the manuscript. Only 12 references are not upto the level of the journal, like "Coatings".
Response: Thank you for your suggestion. More references have been added to the article.
Major issue 2:Comments on the Quality of English Language
Response: Thank you for your suggestion. I have carefully revised the grammar and vocabulary of the article again.
Major issue 3:Still little modifcations are needed.
Response: Thank you for your suggestion. It has been revised.

Round 3
Reviewer 3 Report
Comments and Suggestions for Authors
The manuscript was revised according to the reviewer's comments.
Author Response
Thanks for your comments.
Reviewer 4 Report
Comments and Suggestions for Authors
The following queries have to be addressed by the authors.
1. XRD and SEM images of the alloy before corrosion test is required. Fig. 2 should be revised with better quality. The thickness of the corrosion product layer should be measured in Fig. 6. What are the red marks, d and c indicating in Fig. 6. The authors need to describe the mathematical relation through which the corrosion rate can be calculated as mentioned in Table 7 from the polarization curves by referring the article, Sahu et al. "Effect of Zr content on structure property relations of Ni-Zr alloy thin films with mixed nanocrystalline and amorphous structure" where the authors of the published article have described the how the corrosion rate, Icorr and the Ecorr. can be calculated by extrapolating the polarization curves. It is very important to describe here.
.2. From the XRD plots, it is highly recommended to mention the crystal planes of the peaks. All the cross-sectional SEM images should be magnified with better quality, so that the pores, gap, loose structure should be visible clear (Fig. 4 and 6).
Comments on the Quality of English LanguageSufficient writing skill in the revised version is mandatory.
Author Response

(The authors gave the same response as above.)

Round 4
Reviewer 4 Report
Comments and Suggestions for Authors
Accepted for publication.